# Correlation between Generic and Disease-Specific Quality of Life Questionnaires in Patients with Venous Ulcerations: A Cross-Sectional Study Carried out in a Primary Health Care Setting in Brazil

**DOI:** 10.3390/ijerph20043583

**Published:** 2023-02-17

**Authors:** Dalyanna Mildred de Oliveira Viana Pereira, Adriana Catarina de Souza Oliveira, Severino Azevedo de Oliveira Júnior, Maria Angélica Gomes Jacinto, Alessandra Justino Dionísio, Alana Ellen Oliveira Lima, Bruno Araújo da Silva Dantas, Silvana Loana de Oliveira Sousa, Carmelo Sergio Gómez Martínez, Gilson de Vasconcelos Torres

**Affiliations:** 1Centro de Ciências da Saúde, Universidade Federal do Rio Grande do Norte (UFRN), Rua Piquia, 7932, Pitimbu, Natal 59067-580, Brazil; 2Facultad de Enfermería, Universidad Católica de Murcia (UCAM), C/Joaquín Blume n3. 1º C, 30008 Murcia, Spain; 3Centro de Ciências da Saúde, Universidade Federal do Rio Grande do Norte (UFRN), Av. Olavo Lacerda Montenegro n. 2685, Parnamirim 59158-400, Brazil; 4Centro de Ciências da Saúde, Universidade Federal do Rio Grande do Norte (UFRN), Av. dos Caiapós n. 121, Bairro Pitimbú, Natal 59067-400, Brazil; 5Departamento de Enfermagem, Universidade Federal do Rio Grande do Norte (UFRN), Rua dos Palmares, 30, Parque das Árvores, Parnamirim 59154-145, Brazil; 6Departamento de Enfermagem, Universidade Federal do Rio Grande do Norte (UFRN), Av. Maria Lacerda Montenegro nº 339, Parnamirim 59152-900, Brazil; 7Faculdade de Ciências da Saúde do Trairi, Universidade Federal do Rio Grande do Norte (UFRN), Rua Alice Azevedo, 30, Natal 59080-015, Brazil; 8Facultad de Medicina, Universidad de Murcia (UM), Campus de El Palmar, Edificio LAIB/DEPARTAMENTAL, 30120 Murcia, Spain; 9Facultad de Enfermería, Universidad Católica de Murcia (UCAM), Calle Orden de Santiago, num 5, Abarán-Murcia, 30550 Murcia, Spain; 10Centro de Ciências da Saúde, Research Productivity Scholarship (CNPQ/PQ1D), Universidade Federal do Rio Grande do Norte (UFRN), Rua das Massarandubas, 292, Nova Parnamirim, Parnamirim 59150-630, Brazil

**Keywords:** Varicose Ulcer, venous insufficiency, Quality of Life, observational study

## Abstract

Venous Ulcers (VU) are a serious health problem that affect the Quality of Life (QoL). They are evaluated by many different scales in the literature. We aimed to analyze the correlation between the Medical Outcomes Short-Form Health QoL (SF-36) and Charing Cross Venous Ulcer Questionnaire (CCVUQ) scales. This is a cross-sectional study conducted in a Brazilian center specializing in chronic VU of the Primary Health Care (PHC) provided to patients with active VU. The general QoL instrument SF-36 and the CCVUQ, specific for people with VU, were used. Spearman’s Rho Test determined the correlation between the variables analyzed. Our sample had a total of 150 patients. We found a direct correlation between the domestic activities division (CCVUQ) aspect and the SF-36 Physical role functioning (strong), and Physical functioning (moderate) domains. The Social interaction division (CCVUQ) aspect presented moderate correlation with the domains of the SF-36 Physical role functioning and Physical functioning. The Vitality domain (SF-36) showed moderate correlation with the aspects of CCVUQ Cosmesis division and Emotional status division. The greatest forces of direct correlation were observed between the physical, functional and vitality aspects of SF-36 with those represented by domestic activities and social interaction in the CCVUQ.

## 1. Introduction

Venous Ulcers (VU) are a serious public health problem due to their prevalence and chronicity and have a great impact on people’s health. They have physical, social and behavioral impacts on people’s lives [1]. They are mainly caused by chronic venous insufficiency, characterized by deficiency in the veins that drives blood circulation from the lower limbs towards the heart [2]. VU affects about 1.5% of adults and 5% of individuals over 65 years of age [3,4].

Their impact is also determined by the socioeconomic profile of the patients, since the incidence of VU has shown an association with gender, age, education and income [5]. Chronic venous diseases are more prevalent in Western countries, where more than 2% of the health budget is for VU treatment [6]. However, a study carried out in Taiwan points out that the weekly costs of treating a VU can reach almost US $300, considering that it can take several years of treatment to achieve the healing of the lesion. During this period, patients usually experience depressive feelings as well as physical and social limitations, which directly impact their Quality of Life (QoL) [7,8].

Quality of Life (QoL) is conceptualized according to the World Health Organization by the individuals’ perception of their position in life, in the context of the culture and value system in which they live in relation to their goals, expectations, standards and concerns [9].

QoL has been evaluated in patients with VU, with significant improvement coming from the successful treatment of wounds and their healing [5]. Thus, the importance of using instruments to measure its impact is understood, as is identifying which of its aspects are most impactful [10]. Among the most used instruments to measure QoL, we highlight in this study the Medical Outcomes Short-Form Health QoL (SF-36), which evaluates 10 aspects based on 36 questions on a Likert scale. The survey generates a total score and addresses domains and dimensions of physical, functional, emotional, and social health as well as pain, vitality, and general health status. The evaluation of each aspect generates a score of 0–100 points in ascending order from the worst to the best QoL [11]. The Charing Cross Venous Ulcer Questionnaire (CCVUQ) is intended to measure the QoL of people with VU, considering some particularities commonly experienced by this group. It generates scores for four QoL divisions: domestic, emotional, social, and aesthetic QoL [12].

The correct and effective identification of problems wITH QoL with the aid of these instruments can determine the success of a treatment or the proper management of VU, with a holistic approach that places the individual as the center of care, respecting their context and individuality [13]. For this reason, Primary Health Care (PHC) is an important environment in this process, since it is configured as a gateway for the treatment of patients with VU in the various health systems in the world [14].

In the literature, it is possible to identify several other instruments that measure QoL, either for specific audiences or in general. Researchers have already pointed out that among the aspects of QoL, there may be different results, even when there are similarities between these tools [10]. However, there are few current studies that assess QoL in people with VU. In the Brazilian scenario, in 2018, one study indicated lower QoL scores in these patients, when compared to Portugal, when physical, social and pain aspects were highlighted [15]. In another study, Brazilians with VU presented great impairment of QoL in all aspects analyzed as compared to those without VU [16].

We understand that identifying these interactions between aspects of instruments applied in the same group of people can help researchers and professionals make decisions about the tool that will be used to evaluate their patients. Different scales can approach the same problem in different ways, considering the relevance of one of its aspects to the detriment of the others. In addition, this study is also justified by the need to fill scientific gaps on content analysis of QoL scales, especially for people with VU.

This study aimed to analyze the correlation between the SF-36 and CCVUQ QoL scales, applied to people with VU treated in Brazilian PHC units. Our hypothesis was that there was a direct correlation from moderate to strong between the variables of the scales analyzed.

## 2. Materials and Methods

### 2.1. Ethical Aspects

The study was approved by the Research Ethics Committee of the Federal University of Rio Grande do Norte, Approval No. 156068, as following the ethical standards in force in Brazil and the parameters established by the Declaration of Helsinki of 1975, revised in 2013. Before starting the research, all participants received guidance regarding the objectives, risks, benefits, and importance of the study and signed the informed consent form as an indication of acceptance of participation. 

### 2.2. Study Design and Location

This is an observational, cross-sectional study with a quantitative approach, carried out in 2020 in a center specializing in chronic VU prevention and treatment, where patients with VU from 29 PHC units in the city of Parnamirim, state of Rio Grande do Norte, Brazil, are referred.

### 2.3. Population and Sample

As criteria for inclusion in the study, participants should be over 18 years of age; be registered in the local PHC; have at least one active VU below the knee and present an Ankle-Arm Index greater than 0.8 and less than 1.3. As exclusion criteria, we adopted the following: having a completely healed VU of mixed or non-venous origin; having been discharged from treatment; in the case of change of address to another city. To consider VU active or healed, we considered the presentation of the medical report that accompanied the patient during treatment. One hundred and three (103) people completed the study. 

### 2.4. Instruments and Variables

To collect sociodemographic, care and health information, we used a form prepared and structured by the researchers themselves, which asked about gender, age group, income, marital status, education, presence of comorbidities, use of medications, and treatment time. 

In order to assess QoL and contemplate the objective of the study, we used two instruments. The SF-36 evaluates eight domains and two dimensions of QoL, with questions on a Likert scale aimed at measuring the general well-being of the participants, their physical and emotional limitations, and expectations about their health, generating a score between 0 and 100; the closer to zero, the worse QoL is considered to be [11]. 

We also used the CCVUQ, a specific questionnaire to assess QoL in individuals with VU, composed of 21 questions that identify four important health divisions: domestic activities division, social interaction division, cosmesis division, and emotional status division. The instrument determines a score between 0 and 100; the closer to zero, the better the individual’s QoL score [12]. All instruments were validated and adapted to the Brazilian Portuguese language [11,17].

### 2.5. Data collection and Availability

Before the beginning of the collections, the researchers received qualification training as interviewers. They were undergraduate and graduate students in nursing, nutrition, and medicine at the Federal University of Rio Grande do Norte. The collections were carried out through face-to-face interviews, in which the researcher read the questions and alternatives to the participants and filled them in according to the responses. The questions were guided by the selected instruments, with an approximate duration of 60 min, in the period between August and October 2020. The entire process was supervised by the research coordinators. The sessions occurred during multiprofessional consultations and changes of dressing of the VU, which were scheduled according to the routine of the institution. No incentives or remuneration were offered to any of those involved in the study, and there was no blinding of participants or researchers. All data were deposited and are available in the Mendeley Data repository, available from the link: https://data.mendeley.com/datasets/h6bb7ckhtb (accessed on 15 January 2023).

### 2.6. Analysis and Data Processing

The data were tabulated and presented in tables with the aid of the Microsoft^®^ Excel 2016 software (Microsoft Corporation, Washington, DC, USA). The Statistical Package for the Social Sciences (SPSS) version 20.0 (IBM, Armonk, NY, USA) was used for the analyses of interest. 

The Kolmogorov–Smirnov test identified the non-normality of the sample. Thus, descriptive analyses of absolute and relative frequencies of the categorical variables of the sociodemographic profile were performed, with their association levels measured by Pearson’s nonparametric Chi-square test. We also extracted the mean, Standard Deviation (SD), and percentiles (25, 50, and 75) of the scores of the QoL scalar variables, with their respective differences measured by Pearson’s Chi-square Test. To evaluate the internal consistency of the QoL scales (SF-36 and CCVUQ), we performed Cronbach’s inferential analysis (α) using the following parameters and classification: 0.30 < α ≤ 0.60 (low); 0.60 < α ≤ 0.75 (moderate); 0.75 < α ≤ 0.90 (high); α > 0.90 (very high). To analyze the correlation (*r*) between the scalar variables of both scales, Spearman’s Rho Test was used. Its classification regarding correlation levels was determined by the following parameters: *r* ≤ 0.29 (weak); 0.29 > *r* ≤ 0.49 (moderate); *r* ≥ 0.50 (strong). The same parameters were used in the analysis of confounding factors when we tested the correlation between the scale variables and the subcategories of sociodemographic variables. The meaning of the correlation was classified considering that the SF-36 scale indicates better QoL when it results in higher scores (increasing direction for better QoL), while the CCVUQ scale presents a decreasing sense to determine better QoL. Therefore, it was termed “direct correlation” when the value of *r* was negative and “opposite correlation” with positive *r*. For all tests, we adopted the margin of error of 5% and Confidence Interval (CI) of 95%, with a significance value of *p* < 0.05 [18,19].

## 3. Results

Though 150 individuals registered in the service were sought, 38 had moved to other locations. Therefore, we obtained an initial sample of 112 participants. Of these, nine were excluded, with four of them having been discharged for cure and five dying in the time interval between the appointment contact and the interviews. Therefore, the final sample was n = 103. The sample selection process is shown in Figure 1.

Table 1 presents the sociodemographic characterization of the participants. In it, we highlight that the majority were female (n = 75/72.8%/*p* < 0.001), had an age equal to or greater than 60 years (n = 64/62.1%/*p* = 0.014), income lower than a minimum wage (n = 85/82.5%/*p* < 0.001), had no occupation or work (n = 82/79.2%/*p*< 0.001), and had low education (n = 82/79.6.5%/*p* < 0.001).

Table 2 shows the scores of the aspects of both instruments that measured QoL and the internal consistency of both, with data ranked from the lowest to the highest mean score (from worst to best QoL in SF-36 and from worst to best in CCVUQ). Given the data extracted with the SF-36, we found lower scores in the domains Physical role functioning (mean = 8.1/SD = 17.8/*p* < 0.001) and Physical functioning (mean = 8.2/SD = 27.0/*p* < 0.001). Among its highest scores were the Mental health domain (mean = 57.4/SD = 8.2/*p* < 0.001) and Vitality (mean = 52.1/SD = 11.9/*p* = 0.002). When applying the CCVUQ, we observed lower scores in the Social interaction division (mean = 58.4/SD = 20.1/*p* < 0.001) and higher scores in the Domestic activities division (mean = 63.4/SD = 20.4/*p* < 0.001). Regarding the internal consistency of the scales, the SF-36 presented moderate classification (α = 0.68), while the CCVUQ exhibited high consistency (α = 0.89).

In Table 3, we present the correlation between the aspects of QoL of both scales, ranked according to the correlation strength. In the analysis, the direct correlation between the Domestic activities division (CCVUQ) aspect and the SF-36 Physical role functioning (*r* = −0.50/*p* < 0.001/strong) and Physical functioning (*r* = −0.31/*p* = 0.001/moderate) domains were highlighted. The Social interaction division (CCVUQ) aspect was correlated with the SF-36 domains Physical role functioning (*r* = −0.43/*p* < 0.001/moderate) and Physical functioning (*r* = −0.42/*p* < 0.001/moderate). The Vitality domain (SF-36), was correlated with aspects of the CCVUQ Cosmesis division (*r* = −0.32/*p* < 0.001/moderate) and Emotional status division (*r* = −0.36/*p* < 0.001/moderate).

We also observed the opposite correlation between the Pain domain and the Domestic activities division (*r* = 0.37/*p* < 0.001/moderate); the Emotional role functioning domain and the Cosmesis division (*r* = 0.45/*p* < 0.001/moderate) and Emotional status division (*r* = 0.49/*p* < 0.001/moderate); as well as the Mental health dimension and the Emotional status division (*r* = 0.30/*p* = 0.002/moderate). However, we emphasize that because it is an opposite correlation, these variables had opposite influence between them, in the sense that when one had good QoL, the other correlated variable had worse QoL. 

To verify possible confounding factors, we repeated the correlation analysis between the scales, but within the context of each of the subcategories of sociodemographic variables. Among all the analyses carried out, we observed that in the Education variable, the correlation between the Vitality domain (SF-36) and Emotional status division (CCVUQ) in the “High school or more” subcategory (n = 21) maintained the same strength (moderate) and direction (direct), but not the same level of significance (*r* = 0.34/*p* = 0.135). A similar behavior was observed in the Age Group variable, in the subcategory “<60 years” (n = 39). In this scenario, the strength of correlation between the Pain domain (SF-36) and Domestic activities division (CCVUQ) changed from moderate to weak, also changing its significance level (*r* = 0.27/*p* = 0.93) and sense (opposite).

The direct and opposite correlations are represented, respectively, in Figure 2 and Figure 3.

## 4. Discussion

Our study pointed out as the main evidence the presence of correlation between the QoL scales analyzed (SF-36 and CCVUQ). Among the aspects evaluated, we observed a direct correlation between the physical, functional, and vitality aspects of the SF-36 and the aspects of social interaction, ability in domestic activities, aesthetics, and emotional state of the CCVUQ. The strongest correlation was found between Domestic activities division and the Physical role functioning domain. There was also an opposite correlation between the emotional, mental, and pain aspects in the SF-36 and the domestic, aesthetic, health, and emotional activities of the CCVUQ. The CCVUQ exhibited greater internal consistency compared to the SF-36.

We highlight that these results offer elements that can help professionals to decide between the use one of the scales or their joint use. This choice will depend on which aspect is being measured in people with VU. Thus, this study also contributes to fill scientific gaps on the subject with a situational diagnosis, which can serve as a baseline for the formulation of care policies for patients with VU in PHC in different sociodemographic contexts. 

Regarding the sociodemographic profile observed in our study, we identified similarities with other studies, which showed a predominance of females, older ages, low education, and low income [5,12]. Researchers have explained that the presence of VU is more frequent in women due to the overload on their venous network, which presupposes fluid retention and, consequently, greater venous return difficulties compared to men [2]. 

Older age is inserted in this context due to the natural deficit of venous reflux observed with the aging process and the chronic diseases prevalent among older people [1,2]. Factors such as education and income can impact the development and evolution of VU, since they directly interfere with the individual’s level of self-care [20] as well as the financial capacity to fund the treatment of injuries, which usually requires substantial investment, depending on the technology used [21,22]. Therefore, we also analyzed whether sociodemographic variables could interfere with the correlations. In this analysis, we identified younger age groups and higher education levels as potential confounding factors.

When performing the descriptive analysis of QoL, we verified that the SF-36 data indicated the Physical role functioning and Physical functioning domains as the worst scores. At CCVUQ, the Domestic activities division stood out. This aspect, in turn, presented the highest direct correlation strength among our findings with the Physical role functioning domain, in addition to a moderate direct correlation with Physical functioning. The latter also showed moderate direct correlation with the Social interaction division (CCVUQ). It is noteworthy that these aspects are evaluated according to the physical situation of the individuals and their ability to perform basic and complex day-to-day activities [11], commonly impaired with advancing age and with the progression of diseases and chronic diseases [23]. Considering that our sample was composed of predominantly older people with chronic lesions, it is worth questioning whether this profile is related to the scores found.

Despite some similarities between the variables measured in both scales, it can be seen that some of the aspects present in the SF-36 are not contemplated separately in the CCVUQ. For example, the pain domain (SF-36) presented significantly low scores among the participants. However, controversial data were observed in this aspect, as an opposite correlation of this aspect was observed with the CCVUQ Domestic activities division, although the questions are similar in both instruments regarding the limitations generated by pain in functional activities and daily life. Nevertheless, when we performed the analysis, considering the subcategories of sociodemographic variables, we noticed that this correlation lost strength and statistical significance among participants under 60 years of age. Authors have pointed out that pain is present in most people with VU and is related to local hypoxia triggered by poor circulation in the affected limb, which can lead to impairments in the mobility of the individuals and their willingness to perform homework and social interaction activities [24,25]. It is important to highlight that in other QoL scales used in people with VU, as well as in the SF-36, pain is one of the aspects evaluated individually; this is the case for the Charing Cross Venous Ulcer Questionnaire, Venous Leg Ulcer Quality of Life (VLU-QoL), Sheffield Preference-based Venous Ulcer Questionnaire (SPVU-5D), and the EQ-5D-5L [10].

Regarding social interaction, we list as variables of both instruments the Social role functioning domain (SF-36) and the Social interaction Division (CCVUQ). Both assess QoL in the sense of participating in activities in family groups, co-workers or friends. However, the SF-36 presents general questions, without exemplifying activities in its questions, while the CCVUQ specifies activities that the individual has performed, such as vacation trips and meetings between friends [11,12]. In this sense, it is important to reflect on whether the writing of the instruments directly interferes with the results; when we individually evaluated each of these aspects, both presented statistical significance in their score distribution. However, the correlation between the variables of the social aspect was weak and opposite, that is, when the QoL measured in one of them was good, in the other, it was poor.

It is known that social relations are an important factor for the understanding of the individuals within their environment and satisfaction with life. Various socialization activities may interfere to a lesser or greater degree in the QoL of the individuals, depending on their representativeness in the context of each person. The individual may eventually understand that activities such as a weekly family reunion or a card game entail a great level of satisfaction. A trip with friends to another city or participation in artistic events may not make much sense to the individual; therefore, the fact of not doing such things does not necessarily imply any interference in QoL [26]. In the context of patients with VU, many report being afraid of hurting someone’s sensitivity or ashamed of exposing their injuries in environments with many people, even when covered with a bandage [7,15]. Therefore, specifying activities to measure the aspect of social interaction can be a controversial and counterproductive tool in evaluation scales, depending on what is desired with its application.

The domains General health perceptions and Vitality are also not aspects represented in the CCVUQ and stood out in the evaluation of the participants. The General health perceptions domain assesses how the individuals felt about their health status in recent weeks and the Vitality domain measures how they feel about their degree of energy, which implies the willingness to perform activities [11]. Although both presented statistical significance in the individualized analysis of the scales, only in the Vitality domain saw a direct and moderate correlation with the aspects of the CCVUQ Cosmesis division and Emotional status division. However, higher levels of education proved to be a potential confounding factor for the correlation between the Vitality and Emotional status divisions, which did not show statistical significance in this scenario. 

In this same context, a Spanish study indicated that the aspects that most interfered with QoL were emotional and aesthetic aspects. However, the authors explained that there was variation in the results when considering the severity and extent of VU and the intensity of pain reported by the participants [27]. It should be noted that the CCVUQ evaluates the aspect of pain, but it is not represented in an individualized domain [12]. Among our findings, the pain domain (SF-36) was correlated with the Emotional status division but at a low level and in opposite directions of the scales (opposite correlation). Nor did we assess the clinical severity of the participants or the extent of their VU.

The Emotional role functioning and Mental health domains of the SF-36 stood out significantly in the individual analysis. Regarding the CCVUQ, in the Emotional status division aspect, no significance was observed. Despite the similarity with the Emotional status division (CCVUQ), the correlation of moderate strength with Emotional role functioning was opposite, which suggests different interpretations of the results for the same aspect as evaluated in both scales. This context differs from a study that compared the EQ-5D-5L and SPVU-5D scales, in which it was observed, for example, that the highest direct correlation strength was between the “Anxiety/depression” aspect of one scale and “Mood” measured in the other [10].

Regarding the Mental health domain, we observed its direct correlation with the aspects of the CCVUQ Social interaction division, Cosmesis division, and Emotional status division, but with low strength. It is observed in the literature that depressive symptoms, anxiety, and other mental and emotional disorders are prevalent in the context of patients with VU [27,28]. This phenomenon is explained by the individual having an injury that exerts physical and functional limitations due to the presence of pain, discomfort, and the need for special care for long periods of time [7]. In addition, depression is also an important factor that can make it difficult to adhere to treatment and delay the healing of the lesion [29]. Thus, it is understood that these are essential aspects for evaluation in the context of QoL of these individuals, but require an accurate approach. Our data suggest that there is no obvious similarity between the evaluations of the mental and emotional elements of both scales.

In general, our findings highlighted that the choice instrument in assessing the QoL in people with VU involves considering the content analysis criteria of the scale used as well as the objective that the researcher or professional proposes. Although the internal consistency of the CCVUQ was greater, it is observed that the SF-36 delimits some aspects more clearly, such as pain, vitality, and separating the physical from the functional aspects. Therefore, it is also pertinent to question whether the smaller number of variables in the CCVUQ favored its greater consistency when performing the test. We believe that the SF-36 scale can offer the assessment of a greater number of aspects in an individualized way. In this way, it is possible to analyze the association between each of them, with possible outcomes. 

As a limitation of our study, we did not reach the sample number indicated by the sample test. Another important point is that the interviews were conducted during consultations, which involved the exposure of the VU of the participants, which may have caused discomfort and trigger memories and feelings and, consequently, an overestimation in the answers provided. In order to reduce the possibilities of possible research biases, the data collection environments were reviewed in order to promote participant privacy as well as a comfortable and welcoming setting. In addition, we carried out analyses of possible confounding factors in order to clarify possible interference in the results found. 

## 5. Conclusions

We conclude that the observed correlations point to convergences and divergences in the behavior of some aspects between both questionnaires. The variables that represented the physical, functional, vitality and domestic activities interacted with each other in the same direction as the scales. This means that when an individual had a high score in one of them, the others behaved in the same way (direct correlation). However, other scale variables, even when they represented similar aspects, showed opposite behaviors (opposite correlation), and this could be noticed in a significant way. This was especially noted in the emotional aspects of the SF-36 and CCVUQ. These interpretations allowed us to conclude that our study hypothesis was accepted.

When comparing the behavior of the scales in the sample, we observed that the internal consistency of the CCVUQ was higher. However, the consistency of the SF-36 was not low and offers numerically more elements that can be evaluated in the context of the QoL of the individuals, especially for patients with VU, which implies greater complexity. In this way, the professionals can choose one of the scales or use the two in a combined way.

The evidence of the correlations found may indicate a path for health professionals and managers in the formulation of interventions or strategies to improve the QoL of people with VU, as it was possible to identify which of its aspects are related each other.

It is suggested to deepen the theme with new studies to improve the understanding of the correlation between both scales in other types of work, such as clinical trials or longitudinal studies. The authors also noticed difficulties in finding updated clinical studies on the use of QoL scales in patients with VU, which can be substantially improved.

## Figures and Tables

**Figure 1 ijerph-20-03583-f001:**
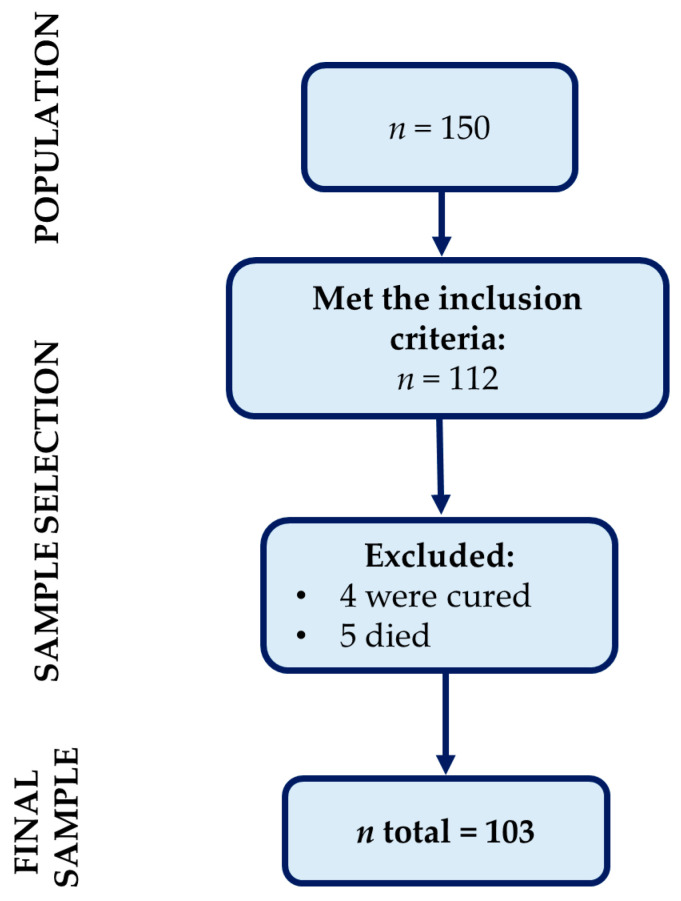
Flowchart of the recruitment process and sample selection.

**Figure 2 ijerph-20-03583-f002:**
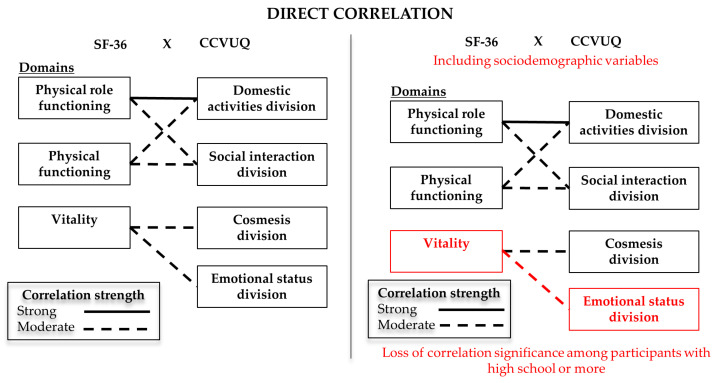
Graphical representation of the direct correlations between the SF-36 and CCVUQ scales.

**Figure 3 ijerph-20-03583-f003:**
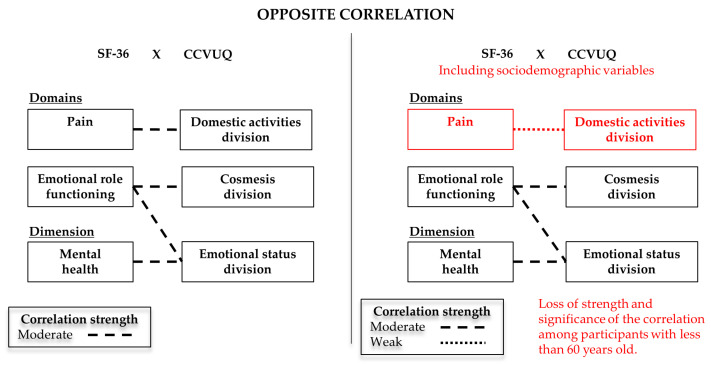
Graphical representation of the opposite correlations between the SF-36 and CCVUQ scales.

**Table 1 ijerph-20-03583-t001:** Sociodemographic characterization of participants.

Sociodemographic Variables	n	%	*p*
Gender			
Female	75	72.8	<0.001
Male	28	27.2	
Age group			
≥60 years	64	62.1	0.014
<60 years	39	37.9	
Marital status			
Single/widowed/divorced	53	51.5	0.078
Married/stable union	50	48.5	
Income			
≤1 minimum wage *	85	82.5	<0.001
>1 and <3 minimum wage	15	14.6	
≥3 minimum wage	3	2.9	
Occupation			
Not active	82	79.6	<0.001
Active	21	20.4	
Education			
Elementary school or less	82	79.6	<0.001
High school or more	21	20.4	

* Minimum wage in Brazil in 2020: 1.045 BRL. ≤: “less than or equal to”; ≥: “greater than or equal to”; <: “less than”; >: “greater than”.

**Table 2 ijerph-20-03583-t002:** Description of the participants’ scores and internal consistency of the QoL scales (SF-36 and CCVUQ).

Scales of QV	Mean (SD)	Percentiles			*p* *	α **
		25	50	75		
SF-36 (n = 103)						
Domains						0.68
Physical role functioning	8.1 (17.8)	0.0	0.0	5.0	<0.001	
Physical functioning	8.2 (27.0)	0.0	0.0	0.0	<0.001	
Pain	25.0 (19.4)	10.0	20.0	40.0	<0.001	
Social role functioning	30.0 (21.9)	12.5	37.5	37.5	<0.001	
Emotional role functioning	48.5 (29.8)	33.3	66.6	66.6	<0.001	
General health perceptions	51.6 (14.2)	45.0	55.0	60.0	<0.001	
Vitality	52.1 (11.9)	45.0	50.0	60.0	0.002	
Mental health	57.4 (8.2)	52.0	56.0	64.0	<0.001	
Total score	35.2 (9.2)	27.9	33.6	40.7	1.000	
Dimensions						
Physical health	29.0 (9.2)	23.0	27.0	33.0	<0.001	
Mental health	48.1 (9.6)	40.7	48.0	40.7	1.000	
CCVUQ (n = 103)						0.89
Cosmesis division	48.6 (18.6)	33.4	48.5	63.5	0.196	
Emotional status division	54.3 (20.1)	37.8	57.5	60.4	0.280	
Social interaction division	58.4 (20.1)	43.6	65.3	71.8	<0.001	
Domestic activities division	63.4 (20.4)	49.9	71.3	83.1	<0.001	
Overall	54.5 (15.6)	44.6	56.0	66.7	1.000	

SD: Standard Deviation; * Pearson’s Chi-square test; ** Cronbach’s consistency: 0.30 < α ≤ 0.60 (low); 0.60 < α ≤ 0.75 (moderate); 0.75 < α ≤ 0.90 (high); α > 0.90 (very high).

**Table 3 ijerph-20-03583-t003:** Correlation between the aspects evaluated in the SF-36 and CCVUQ scales among the participants.

Scales of QV	CCVUQ				
	Domestic Activities Division	Social Interaction Division	Cosmesis Division	Emotional Status Division	Overall
SF-36	*r* * (*p* **)	*r* (*p*)	*r* (*p*)	*r* (*p*)	*r* (*p*)
Domains					
Physical role functioning	−0.50 (<0.001)	−0.43 (<0.001)	−0.15 (0.143)	−0.16 (0.103)	−0.34 (<0.001)
Physical functioning	−0.31 (0.001)	−0.42 (<0.001)	−0.19 (0.056)	−0.20 (0.039)	−0.30 (0.002)
Vitality	−0.29 (0.003)	−0.29 (0.003)	−0.32 (<0.001)	−0.36 (<0.001)	0.39 (<0.001)
Mental health	0.06 (0.530)	−0.21 (0.030)	−0.27 (0.006)	−0.27 (0.005)	−0.27 (0.006)
Pain	0.37 (<0.001)	0.29 (0.003)	0.16 (0.098)	0.21 (0.029)	0.30 (0.002)
Emotional role functioning	0.29 (0.003)	0.19 (0.058)	0.45 (<0.001)	0.49 (<0.001)	0.47 (<0.001)
Social role functioning	0.20 (0.043)	0.21 (0.036)	0.13 (0.193)	0.16 (0.108)	0.23 (0.020)
General health perceptions	0.25 (0.010)	0.30 (0.002)	0.15 (0.137)	0.16 (0.111)	0.26 (0.007)
Total score	0.17 (0.088)	0.03 (0.740)	0.12 (0.229)	0.17 (0.090)	0.17 (0.079)
Dimensions					
Physical health	−0.10 (0.300)	−0.17 (0.088)	−0.22 (0.025)	−0.21 (0.036)	−0.19 (0.055)
Mental health	0.28 (0.004)	0.18 (0.062)	0.26 (0.008)	0.30 (0.002)	0.33 (0.001)

* Correlation levels: *r* ≤ 0.29 (weak); 0.29 > *r* ≤ 0.49 (moderate); *r* ≥ 0.50 (strong).** Spearman’s Rho test; *p*-value for Spearman’s coefficient.

## Data Availability

The data presented in this study are openly available in Mendeley Data at doi:10.17632/h6bb7ckhtb.1, available at the link: https://data.mendeley.com/datasets/h6bb7ckhtb (accessed on 15 January 2023).

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
