# Peer review of "Correlation between Generic and Disease-Specific Quality of Life Questionnaires in Patients with Venous Ulcerations: A Cross-Sectional Study Carried out in a Primary Health Care Setting in Brazil"

_ijerph, 2023, doi:10.3390/ijerph20043583_

Round 1

Reviewer 1 Report

Manuscript under review assesses the quality of life for ulcer patients. Study is based in Brazil and carried out thru correlation analysis between QoL Medical Outcomes Short-Form Health QoL (SF-36) and Charing Cross Venous Ulcer questionnaire scales. Literature review is limited but seems enough for a study with specific target. Sampling size and method is appropriate. Statistical analysis seems to be carried out properly as well. Supplementary material provided by the authors is nice.

I would only suggest the authors to present some arguments for healthcare managers/practitioners on utilizing the outcomes of the study. How strong correlations can be used to improve QoL for ulcer patients?

Reviewer 3 Report

Dear authors,

You raised an important issue of perfect-matching quality-of-life questionnaires for patients suffering from VU. 

There are some major questions mainly regarding the methodology that needs to be answered and suggestions that need to be addressed before making a final decision on your manuscript. 

1) To descriptive title. I would suggest: "Correlation between generic and disease-specific quality of life questionnaires in patients with venous ulcerations"

2) Line 51-52: This sentence is too pop-science simplified phrase. Remove it or re-write it describing in a scientific way the pathomechanism leading to veins insufficiency. 

3) Line 55-56: The sentence does make sense. First of all, you mixed the causes (HT, DM) with ailments (slow healing). Second of all, DM and HT are not the main reasons for VU. 

4) References: There are multiple references not related to the information given. Some of them are misused, and some are related to information cited by other articles but the original data. Refer to the original source, do not re-refer. 

5) 2.3. Population and sample - first sentence: This is unnecessary duplication of information given in 2.2 paragraph - remove. 

6) Line 123-124: How were the participants picked if they were not just consecutive patients meeting inclusion and exclusion criteria? What was the pattern? 

7) Line 124-128: This is not the proper methodology of power and sample size calculation. This paragraph should be removed. 

8) Inclusion/exclusion criteria: How did you assess what is the cause of an unhealing wound? Was there a Doppler ultrasound examination of lower limb veins performed? The inclusion criteria are not solid or exact enough to exclude the patients with wounds of other origins based only on those mentioned.  

9) Line 149-150: Reference to validation needed.

10) Data collection: To be precise - the researches were reading the questions or the patients were filling out the paper/electronic questionnaire by themselves supervised by researchers? This needs to be specified. 

11) In the discussion, you mention that the study results may benefit to make decisions on the proper choice of tool for QoL assessment in VU patients, but there is no information on what would be your recommendation on which patients. You need to include some conclusion and a real take-home message. 

12) Line 268-269: You show a great bias in gender, income, education status and age, but did not include it in any of the calculations and analysis. In my opinion, it would be beneficial if you perform the subgroup analysis and show if the mentioned above factors somehow influence not the answers themselves, but the relationship between the questionnaires. Without it, the study does not bring any new significant information to already examined and previously published results.  

13) Line 365-366: If you try to compare the tools in the aspect of their efficacy and suitability the chosen by you cross-sectional study design is necessary and I would not take it as a study limitation, but its strength. 

14) Conclusions paragraph: This paragraph should contain conclusions and a take-home message, not a summary of the results. It needs to be rewritten and focused on what the results mean instead of what they are. 

Kind regards

Round 2

Reviewer 3 Report

Dear Author,

Thank you for your effort and for improving the manuscript. I agree with the corrections and improvements that you introduced in the article. Unfortunately, at the moment the article does not bring much new merit into the studied subject, it is rather a confirmation of what was already published. I would once again encourage you to perform additional analysis of the confounding factors. 

Your sincerely
